# Enhancing Wound Healing and Anti-Inflammatory Effects by Combination of CIGB-258 and Apolipoprotein A-I against Carboxymethyllysine Toxicity in Zebrafish: Insights into Structural Stabilization and Antioxidant Properties

**DOI:** 10.3390/antiox13091049

**Published:** 2024-08-28

**Authors:** Kyung-Hyun Cho, Yunki Lee, Sang Hyuk Lee, Ji-Eun Kim, Ashutosh Bahuguna, Maria del Carmen Dominguez-Horta, Gillian Martinez-Donato

**Affiliations:** 1Raydel Research Institute, Medical Innovation Complex, Daegu 41061, Republic of Korea; 2Center for Genetic Engineering and Biotechnology, Ave 31, e/158 y 190, Playa, La Havana 10600, Cuba; mcarmen.dominguez@cigb.edu.cu (M.d.C.D.-H.);

**Keywords:** apolipoprotein A-I (apoA-I), CIGB-258 (Jusvinza^®^), high-density lipoproteins (HDL), carboxymethyllysine (CML), zebrafish, interleukin-6

## Abstract

CIGB-258 is known to exert anti-inflammatory activity via structural stabilization of apolipoprotein A-I (apoA-I) and functional enhancement of high-density lipoproteins (HDL) against acute toxicity of carboxymethyllysine (CML). The co-presence of CIGB-258 in reconstituted HDL (rHDL) formed larger rHDL particles and enhanced anti-inflammatory activity in a dose-dependent manner of apoA-I:CIGB-258, 1:0, 1:0.1, 1:0.5, and 1:1 of molar ratio, in the synthesis of the rHDL. However, no study has evaluated the enhancement of HDL functionality by the co-presence of lipid-free apoA-I and CIGB-258. The present study was therefore designed to compare the structural stabilization and functional improvement of HDL in the presence of lipid-free apoA-I and CIGB-258 in molar ratios of 1:0, 1:0.1, 1:0.5, and 1:1 within both HDL_2_ and HDL_3_. As the concentration of CIGB-258 increased, it effectively inhibited the cupric-ion-induced oxidation of HDL, thereby safeguarding apoA-I from proteolytic degradation. Additionally, the wound-healing activity of zebrafish was significantly (*p* < 0.01) enhanced by the co-addition of apoA-I:CIGB-258 (1:1) up to 1.6-fold higher than apoA-I alone (1:0) under the presence of CML. ApoA-I:CIGB-258 (1:1) treatment exhibited the lowest apoptosis and production of reactive oxygen species against CML-induced damage in the wound site. Also, an increase in wounded tissue granulation and epidermis thickness was observed with increasing concentration of CIGB-258 during 48 h post-treatment via the healing process. Intraperitoneal injection of apoA-I:CIGB-258 mixture remarkably ameliorated the acute paralysis and restored zebrafish swimming ability impaired by the acute toxicity of CML. The increase of CIGB-258 content, especially co-injection of apoA-I:CIGB-258 (1:1), leads to a significant 2.3-fold (*p* < 0.001) and 4.1-fold (*p* < 0.001) higher zebrafish survivability and recovery of swimming ability, respectively, than those of CML-control. In the apoA-I:CIGB-258 (1:1) group, neutrophil infiltration and interleukin (IL)-6 production was lowest in the hepatic tissue with the least cellular damage and apoptosis. Additionally, the group treated with apoA-I:CIGB-258 (1:1) demonstrated the lowest plasma levels of total cholesterol (TC) and triglycerides (TG), along with minimal damage to the kidney, ovary, and testicular cells. Conclusively, co-treatment of CIGB-258 with apoA-I effectively mitigated acute inflammation in zebrafish, safeguarded vital organs, structurally stabilized apoA-I, and enhanced HDL functionality.

## 1. Introduction

High-density lipoproteins (HDL) and apolipoprotein A-I (apoA-I) are well recognized for their antioxidant, anti-glycation, and anti-inflammatory properties in the bloodstream, which help to mitigate the risk of type 2 diabetes, rheumatoid arthritis, and sepsis [1,2,3]. Apolipoprotein A-I (apoA-I) is the major protein present in HDL, accounting for about 70% of the protein content of HDL [4,5]. During the synthesis of HDL, lipid-free apoA-I interacts with phospholipids and cholesterol, and later with additional functional components such as paraoxonase (PON-1), resulting in the functional HDL particle [4]. Usually, two to five molecules of apoA-I are present in the HDL [5]. In addition to cardiovascular disease, HDL-cholesterol (HDL-C) reduced levels have been linked to increased incidences of pro-inflammatory and infectious diseases [6,7]. Also, impaired HDL functionality and reduced apoA-I levels are acknowledged risk factors for autoimmune diseases, including systemic lupus erythematosus (SLE), rheumatoid arthritis (RA), psoriasis, and atopic dermatitis [8,9,10]. It has been well known that apoA-I possesses potent anti-inflammatory and anti-tumorigenic activity [11,12] via preventing endotoxin recognition by a toll-like receptor (TLR)-4 and inhibiting pro-inflammatory pathways [13]. An elevated expression of apoA-I was observed to reduce ovarian cancer progression and suppress hepatocellular carcinoma proliferation through the downregulation of the mitogen-activated protein kinase (MAPK) pathway [12,14]. Furthermore, lower expression of apoA-I or dysfunctional apoA-I was associated with the progression of various types of cancer via impairment of cholesterol trafficking and dysregulation of innate immunity [15].

CIGB-258, Jusvinza^®^, is a 3 kDa peptide derived from heat shock protein 60 (HSP60) to exert potent anti-inflammatory activity to suppress cytokine storm via a decrease of interleukin (IL)-6 and tumor necrosis factor (TNF)-α [16,17]. Additionally, CIGB-258 has demonstrated substantial efficacy as a therapeutic agent across various studies, especially showing great promise in treating rheumatoid arthritis. This includes notable results in preclinical trials conducted on rats [18] and mice [19], as well as clinical studies involving human subjects [20]. It has been established that CIGB-258 could protect apoA-I from oxidation and glycation, stabilizing apoA-I and HDL structure against denaturation and degradation under both lipid-bound and lipid-free states [21,22]. An oxidative stress treatment of ferrous ions into human HDL_3_ caused severe degradation of apoA-I and protein aggregation; however, co-treatment of CIGB-258 protected the apoA-I from the proteolytic attack [17]. Treatment of carboxymethyllysine (CML) into HDL and apoA-I, as a glycation stress, induced proteolytic degradation and exposure of intrinsic tryptophan (Trp) to hydrophilic phase via denaturation of apoA-I. The stability of apoA-I/HDL, concerning the Trp residue, is crucial for understanding protein structure and dynamics [23]. Trp residues in apoA-I are positioned within the 3D structure, where they interact with the acyl radicals of phospholipids. In contrast, charged amino acids in apoA-I interact with the polar residues of phospholipids in the aqueous environment [24], thereby stabilizing the apoA-I/HDL complex. Exposure of intrinsic Trp residues to the aqueous phase under stress conditions signifies structural destabilization, compromising HDL functionality. However, co-presence of CIGB-258 protected the HDL and apoA-I from the degradation and denaturation via stabilization of tertiary structure of apoA-I [25].

Interestingly, the protection of apoA-I and enhancement of HDL functionality was highly dependent on the increase of CIGB-258, up to a molar ratio of apoA-I:CIGB-258, 1:0, 1:0.1, 1:0.5, and 1:1, contents both in lipid-free and lipid-bound state. There has been synergistic anti-inflammatory activity of apoA-I and CIGB-258 in reconstituted HDL (rHDL) against acute toxicity of CML in zebrafish adults and embryos via enhancement of rHDL particle size and protection of tertiary structure [21]. Similarly, lipid-free apoA-I and CIGB-258 also displayed synergistic anti-inflammatory activity against the CML-induced acute death of zebrafish embryos through anti-glycation and antioxidant activities [22]. The low-density lipoprotein (LDL) is another important lipoprotein of blood that typically contains apo-B [26], unlike the apoA-I that is exclusively present in the HDL. The apo-B of LDL is vulnerable to oxidation against a variety of stimuli, such as cupric ions [27]. The oxidized LDL (oxLDL), easily recognized by the scavenger receptor of macrophage, leads to the formation of foam cells and atherosclerotic plaque generation [28]. While the apoA-I:CIGB-258 mixture has demonstrated significant antioxidant properties in preventing LDL oxidation induced by cupric ions, its effect on the oxidation of HDL_2_ and HDL_3_ has not been documented. HDL_2_ and HDL_3_, both are apoA-I containing, share similarities in their structure and function. However, HDL_2_ represents a more advanced form of HDL, enriched with cholesterol and thus larger particle size. In contrast, HDL_3_ is the primitive form of HDL, which, upon accumulating cholesterol, is transformed into HDL_2_ [4]. The protein-to-lipid ratio differs, with HDL_2_ exhibiting a 40:60 ratio and HDL_3_ a 55:45 ratio [4]. Beyond these quantitative differences, HDL_2_ and HDL_3_ also vary in the types of protein and lipids they contain [4]. Notably, HDL_3_ tends to have a higher content of antioxidant enzymes, such as paraoxonase (PON) [4].

Safeguarding HDL from oxidative and glycation stress is crucial, as it preserves HDL’s functionality, particularly its positive impact on wound healing. Wound healing is a highly organized event that involves four continuous phases: establishment of homeostasis, inflammation initiated with an accumulation of cytokine-producing leukocytes, cleaning of the damaged area, and remodeling of the extracellular matrix [29]. Recent studies suggest that HDL has a positive healing impact on chronic wounds via the modulation of inflammation [30].

Expanding on the insights of the earlier research, the current study investigated the synergistic interaction of apoA-I:CIGB-258 in safeguarding human HDL_2_ and HDL_3_ against oxidative stress, focusing on improvement in particle morphology, shape, size, and ferric ion reduction capacity. Additionally, physiological assessments were conducted to evaluate the wound healing and anti-inflammatory activities of the apoA-I:CIGB-258 mixture in the presence of CML in adult zebrafish. The zebrafish was selected as a model organism in the present study due to its high genome similarity with humans [31], which makes it an ideal model organism for preclinical studies [32]. Similar to humans, zebrafish possess several vital receptors and enzymes involved in lipid metabolism [33] that render zebrafish a preferred model organism for lipid research. To note, the zebrafish skin morphology mimics the human skin that follows a nearly similar wound healing process such as inflammation, re-epithelization, granulation tissue generation, and remodeling, thus making zebrafish an ideal cutaneous wound healing model for the preclinical studies [34,35]. Consequently, zebrafish were employed in the present study to assess the therapeutic effects of apoA-I in combination with CIGB-258 against CML-induced chronic wounds and dyslipidemia.

## 2. Materials and Methods

Jusvinza^®^ (GIGB-258), a 27 amino acid recombinant peptide from heat shock protein (HSP) 60, was complementary provided by the Center for Genetic Engineering and Biotechnology (CIGB), Havana, Cuba, for the laboratory research. Unless specified, all other chemicals are of analytical grade and used as supplied. Appendix A contains a detailed specification of all the chemicals and reagents used.

### 2.1. Isolation of High-Density Lipoproteins (HDL)

The high-density lipoproteins (HDL_2_ and HDL_3_) were isolated from the human blood following the earlier described method [17,36]. Briefly, 10 mL of the blood was centrifuged to obtain plasma. The 3 mL of plasma was processed for the sequential density gradient ultracentrifugation in the density range of <1.019 g/mL (for VLDL), 1.019 < d < 1.063 g/mL (for LDL), 1.063 < d < 1.125 g/mL (for HDL_2_), and 1.125 < d < 1.225 g/mL (for HDL_3_) prepared by sodium bromide (NaBr) following 24 h centrifugation at 100,000× *g*. The isolated HDL_2_ and HDL_3_ were extensively dialyzed overnight (~24 h) using Tris-buffered saline (pH 8.0) to remove the salts and other small molecules from plasma, such as uric acid and small peptides. The dialysis was performed employing a cellulose membrane (Sigma Cat # D9277, molecular cut-off = 14 kDa). During the dialysis, the TBS buffer was periodically changed after ~5 h intervals. The dialyzed samples were electrophoresed at 15% SDS-PAGE to examine the purity of HDL for any additional plasma proteins, such as albumin.

### 2.2. Isolation Apolipoprotein A-I (apoA-I)

The apoA-I was extracted from HDL using organic solvent extraction and fast protein liquid chromatography, following the methodology described earlier [37]. The isolated apoA-I purity was assessed by sodium dodecyl sulfate-polyacrylamide gel electrophoresis (SDS-PAGE).

### 2.3. Preparation of apoA-I and CIGB-258 Mixture

The apoA-I was blended with CIGB-258 at a molar ratio of 1:0, 1:0.1, 1:0.5, and 1:1 to prepare the four combinations of apoA-I with CIGB-258. The desired ratio of apoA-I and CIGB-258 was amalgamated in the phosphate-buffered saline (PBS, pH 7.0) followed by 1 h vertexing at 37 °C to make the homogenous suspension.

### 2.4. Oxidation of HDL

The effect of apoA-I:CIGB-258 to prevent CuSO_4_ medicated HDL oxidation was assessed by mixing 50 μL of HDL_2_ or HDL_3_ (2 mg/mL) with 5 μL of 0.1 μM CuSO_4_ in the presence and absence of apoA-I:CIGB-258 at the molar ratio of 1:0, 1:0.1, 1:0.5, and 1:1. The 50 μL of HDL_2_ or HDL_3_ (5 μg/mL) suspended in PBS in the absence of CuSO_4_ and of apoA-I:CIGB-258 was prepared that served as a control. The reaction mixture was incubated at room temperature for 30 min following SDS-PAGE (15%). The separated bands were visualized by staining with 1.25% commissive brilliant blue (CBB). The intensity of the separated bands was examined using Image J software (http://rsb.info.nih.gov/ij, accessed on 16 January 2023, version 1.53).

The extent of oxidation was also analyzed through the thiobarbituric reactive substance (TBARS) assay, with malondialdehyde (MDA) as a serving reference, as the protocol described previously [38].

### 2.5. Transmission Electron Microscopic (TEM) Examination of HDL

The morphological changes of HDL_2_ and HDL_3_ treated with CuSO_4_ with and without apoA-I:CIGB-258 (at molar ratio 1:0, 1:0.1, 1:0.5, and 1:1) were examined by TEM as previously described [39]. In brief, 5 μL of HDL_2_/HDL_3_ reaction mixture was mixed with an equal value of phosphotungstic acid (2%, pH 4) and spread over a 200-mesh copper grid following 2 h incubation at 50 °C and subsequent visualization under TEM (Hitachi-HT7800, Tokyo, Japan) at 150 k magnification at an acceleration voltage of 80 kV.

### 2.6. Ferrous Ion Reduction Ability

Ferrous ion reduction ability (FRA) of HDL_2_ and HDL_3_ with CuSO_4_ in the presence and absence of apoA-I:CIGB-258 was examined as a previously described [40]. In brief, 10 μL of HDL_2_ and HDL_3_ (2 mg/mL) was mixed with CuSO_4_ (0.1 μM, final) with and without 2 μL of apoA-I:CIGB-258 (at molar ratio 1:0, 1:0.1, 1:0.5, and 1:1) followed by the addition 180 μL of freshly prepared FRA reagent. After 60 min incubation at room temperature (RT), the absorbance at 593 nm was recorded.

### 2.7. Zebrafish Maintenance

The wild zebrafish (*Danio rerio*) AB strain was procured from the local market and maintained at 28 °C water temperature with a 14 h/10 h light and dark cycle following the standard guidelines adopted by the Animal Care Committee and the Use of Raydel Research Institute (RRI-20-003, approval code). A normal tetrabit (Gmbh D49304, Melle, Germany) was used to feed the zebrafish.

### 2.8. Wound Healing Activity

Zebrafish (n = 60) were anaesthetized using 0.1% phenoxyethanol, and a circular cutaneous wound (2 mm) was made on the dorsal surface using a biopsy punch (Kai Industries Co., Ltd., Oyana, Japan). The zebrafish were randomly allocated in 6 groups (n = 10/group). The cutaneous wound of zebrafish in the PBS group was treated with 1 μL of PBS. The cutaneous wound of zebrafish in the CML group was treated with 1 μL CML (25 μg), while the zebrafish in CML+ apoA-I:CIGB-258 molar ratio 1:0, 1:0.1, 1:0.5, and 1:1 group was treated with 1 μL of apoA-I:CIGB-258 at 1:0, 1:0.1, 1:0.5, 1:1 molar ratio containing CML (equivalent to 25 μg), respectively. Following the given treatment, zebrafish were transferred into their respective tanks, and the wound was monitored periodically under a microscope for up to 48 h.

### 2.9. Visual Observation and Histological Analysis

Wounded areas of the zebrafish across all the groups were visualized at 0, 2, 6, 24, and 48 h after being stained with methylene blue [34,41]. A 2 μL of methylene blue (0.1%) was spread over the wounded area; after 1 min incubation, the injured site was rinsed three times with water, and the wounded area was visualized under the microscope. The wounded area (blue stained) was quantified using Image J software (http://rsb.info.nih.gov/ij/ version 1.53r, accessed on 16 January 2023) and the percentage wound healing was calculated by comparing the wounded area appeared at 2, 6, 24, and 48 h with respect to the area quantified at 0 hr.

At 48 h, four fishes from each group were sacrificed, and the tissue of the wound area was preserved in 10% formalin following alcohol dehydration and tissue sectioning (5 μm thick). The morphological changes in the tissue section were examined by hematoxylin and eosin (H&E) staining [42]. In addition, dihydroethidium (DHE) [43] and acridine orange (AO) [44] fluorescent staining was performed in the tissue section. In brief, tissue (5 μm thick) was covered with 0.25 mL of DHE (30 μM) and AO (5 μg/mL) fluorescent stain. Following 30 min incubation in the dark, the stained section was visualized under fluorescent microscopy at wavelength 585 nm (excitation) and 615 nm (emission) for the DHE and 505 nm (excitation) and 635 nm for the detection of the AO-stained area.

### 2.10. Carboxymethyllysine (CML) Induced Acute Inflammation in Zebrafish

Adult zebrafish (n = 240) were randomly segregated into six groups (n = 40/group). Zebrafish in the PBS group were injected with 10 μL PBS, while 10 μL PBS containing 250 μg CML was injected in the CML group. The zebrafish in apoA-I:CIGB-258 1:0, 1:0.1, 1:0.5, 1:1 group were injected with 10 μL of apoA-I:CIGB-258 at molar ratio 1:0, 1:0.1, 1:0.5, 1:1 containing 250 μg CML, respectively. All injections were carefully carried out with the same injection volume at the same injection site and a nearly similar injection depth using a graduated micro needle to avoid injection errors. Zebrafish across all the groups were monitored for swimming activity and survivability during 60 min post-treatment.

### 2.11. Blood Analysis and Tissue Collection

Zebrafish across all the groups were sacrificed after 60 min post-treatment, and immediately, blood was collected in the tubes pre-rinsed with EDTA. Simultaneously, the liver and kidney were surgically excised and preserved in 10% formalin. The blood sample was processed for the quantification of total cholesterol (TC), triglycerides (TG), high-density lipoprotein cholesterol (HDL-C), aspartate transaminase (AST) and alanine transaminase (ALT) using the commercial assay kits following the instruction of the manufacturers. A detailed procedure is outlined in Appendix A.

### 2.12. Histological and Immunohistochemical Analysis (IHC)

Tissues from the liver and kidney were sectioned (5 μm thick) and processed for the hematoxylin and eosin (H&E) staining following the process mentioned in Section 2.9. For the oil red O (ORO) staining, the tissue section was flooded with the ORO stain and kept at 60 °C [45]. After 30 min incubation, the tissue section was rinsed and counter-stained with hematoxylin for 30 s, followed by a thorough washing with water and visualization under the microscope.

For the IHC liver section (5 μm thick) was incubated overnight with 200×diluted anti-IL-6 antibodies (ab9324) and subsequently developed with EnVison + System horseradish peroxidase (HRP) labelled polymer kit (K4001, Dako, Glostrup, Denmark) containing 1000×diluted HRP-conjugated anti-anti IL-6 antibodies. The section was visualized under a microscope, and images were captured by a Motic cam2300 CCD camera embedded with a stereomicroscope (Motic SMZ 168; Hong Kong). The captured images were processed for the red conversion of the IHC stained area (brown color) at the brown color threshold value (20–120) using Image J software (http://rsb.info.nih.gov/ij/, version 1.53r, accessed on 16 January 2023) to enhance the clarity of the IL-6 stained area.

### 2.13. Dihydroethidium (DHE) and Acridine Orange (AO) Fluorescent Staining

The tissue section (5 μm thick) was fluorescent stained with DHE and AO fluorescent stain to examine the reactive oxygen generation (ROS) and apoptosis, respectively, following the methodology outlined in Section 2.9.

### 2.14. Statistical Analysis

The values obtained from the triplicate experiments were depicted as mean ± SEM. The SPSS software version 29.0 (Chicago, IL, USA) was used to conduct a one-way analysis of variance (ANOVA) followed by Tukey’s post hoc investigation to establish the statistical difference between the groups.

## 3. Results

### 3.1. Higher Content of CIGB-258 More Protected HDL_2_ from Oxidation

As shown in Figure 1A, treatment of cupric ion into native HDL_2_ (lane N) caused a decrease of apoA-I band intensity due to proteolytic degradation of apoA-I (lane O), suggesting that structural stability of HDL_2_ was susceptible to the attack of the oxidative stress posed by cupric ions (final 1 μM). The co-treatment of apoA-I and CIGB-258 induced stronger band intensity of apoA-I in HDL_2_ in a dose-dependent manner of CIGB-258. Especially at 1:0.5 and 1:1 molar ratio of apoA-I and CIGB-258 (lane 3 and 4) prevent the cupric ions mediated degradation manifested by 15.4% to 26.2% enhancement of HDL_2_ associated apoA-I band intensity, respectively as compared to the band intensity of apoA-I observed in HDL_2_ treated with cupric ion (Lane O).

Quantification of oxidized species using TBARS assay revealed that the cupric ion oxidized HDL_2_ showed a 6.5-fold higher MDA level than that of native HDL_2_ (Figure 1B), suggesting that the proteolytic degradation of apoA-I is associated with the production of oxidized species. However, co-treatment of the apoA-I:CIGB-258 mixture caused a significant reduction of the cupric ion-induced MDA production in HDL_2_ by 33% (*p* < 0.01), 42% (*p* < 0.001), and 49% (*p* < 0.001) at the apoA-I and CIGB-258 content of molar ratio 1:0.1, 1:0.5, and 1:1, respectively.

### 3.2. Recovery of HDL_2_ Particle Morphology and Antioxidant Ability

As shown in Figure 2A, native HDL_2_ (photo a) showed a distinct spherical shape of typical HDL particle morphology with the regular size around 137 ± 6 nm^2^ particle distribution; however, cupric ion treatment into HDL_2_ (photo b) caused ambiguous and distorted particle morphology with an irregular particle size of 75 ± 8 nm^2^ (Figure 2B) that occasionally aggregated (as indicated by red arrow). On the contrary, the co-addition of apoA-I:CIGB-258 into the oxidized HDL_2_ improved the particle morphology with the enlargement of particle size in a dose-dependent manner of apoA-I:CIGB-258 ratio. The highest apoA-I:CIGB-258 content (1:1) treated HDL_2_ (photo f) showed the clearest and most distinct morphology compared with 1:0.1 and 1:0.5 molar ratio treated HDL_2_ as shown in photos d and e, respectively. A marked ~1.7-fold enhancement in the HDL_2_ particle size was observed in the 1:0.5 and 1:1 ratio apoA-I:CIGB-258 treated group compared to the particle size of only cupric ion treatment into HDL_2_ (Figure 2A,B).

As shown in Figure 2C, the native HDL_2_ exhibited the highest ferric ion reduction ability (FRA), while oxidized HDL_2_ showed the least FRA ability, around 37% less than native HDL_2_. Interestingly, the co-addition of apoA-I:CIGB-258 into the oxidized HDL_2_ enhanced the FRA in a dose-dependent manner of CIGB-258; although the co-addition of apoA-I:CIGB-258 at 1:0 and 1:0.1 molar ratio exhibited no significant difference. The co-addition of apoA-I:CIGB-258 at 1:0.5 and 1:1 induced remarkable elevation of FRA up to 33% (*p* < 0.01) and 60% (*p* < 0.001) higher than that of the FRA of oxidized HDL_2_.

### 3.3. The Higher CIGB-258 Content the More Protection of HDL_3_ from the Oxidation

As shown in Figure 3A, cupric ion treatment into native HDL_3_ (lane N) caused proteolytic degradation of apoA-I, evidenced by the almost disappearance of the protein band (lane O), suggesting that the vulnerability of HDL_3_ toward the oxidation. The co-treatment of apoA-I:CIGB-258 prevented the cupric ion-mediated proteolytic degradation HDL_3_ in a dose-dependent manner of the CIGB-258 content. Although, a visible effect of apoA-I alone (1:0) was also observed to prevent the cupric ion-mediated proteolytic degradation HDL_3_ as evidenced by the 2.5-fold higher band intensity (lane 1) than oxHDL_3_. A profound HDL_3_ preventive effect was observed with the enhancement of CIGB-258 content in apoA-I. At the 1:0.1, 1:0.5, and 1:1 molar ratio of apoA-I:CIGB-258 a 3.0-fold, 3.8-fold, and 3.9-fold higher apoA-I band intensity was observed, compared to the oxHDL_3_, suggesting that synergistic activity between apoA-I and CIGB-258 to protect HDL_3_.

Quantification of oxidized species revealed that oxidized HDL_3_ showed the highest MDA level, around 5.8-fold higher than that of native HDL_3_ (Figure 3B), signifying that production of MDA was associated with proteolytic degradation of apoA-I in the HDL_3_. Interestingly, the co-addition of apoA-I:CIGB-258 resulted in the prevention of MDA production in HDL_3_. A dose-dependent effect of apoA-I:CIGB-258, at a molar ratio 1:0, 1:0.1, 1:0.5, and 1:1, was noticed to prevent MDA production evident by a significant 17% (*p* < 0.05), 31% (*p* < 0.001), 46% (*p* < 0.001), and 51% (*p* < 0.001) lower MDA level, respectively, compared to the oxidized HDL_3_.

### 3.4. Morphological Change of HDL_3_ by Oxidation and Prevention

As shown in Figure 4A, native HDL_3_ (photo a) showed a distinct and clear shape with regular particle distribution. In contrast, oxidized HDL_3_ (photo b) exhibited almost a disappearance of the original shape with frequent particle aggregated patterns and 43% (*p* < 0.001) less particle size than native HDL_3_ (Figure 4B). Although, co-addition of apoA-I alone (1:0) was ineffective in improving the cupric ion distorted HDL_3_ particles morphology. However, the co-addition of apoA-I:CIGB-258 resulted in the prevention of HDL_3_ particle morphology impaired by the exposure of cupric ions. The more distinct particle morphology with enlargement of particle size, around 37% (*p* < 0.001), 55% (*p* < 0.001), and 65% (*p* < 0.001) enhancement of HDL_3_ particle size was observed in response to the treatment of apoA-I:CIGB-258 at a molar ratio of 1:0.1, 1:0.5, and 1:1, respectively, than the particle size of ox HDL_3_ (Figure 4B).

The antioxidant ability suggested a 26% (*p* < 0.001) reduced FRA of oxidized HDL_3_ as compared to the native HDL_3_ (Figure 4C). The co-addition of apoA-I alone did not prevent the loss of FRA activity of HDL_3_ induced by the exposure of cupric ions; however, the co-addition of apoA-I:CIGB-258 enhanced the FRA ability with increase content of CIGB-258 as manifested by around 18% (*p* < 0.01), 23% (*p* < 0.01), 51% (*p* < 0.001) higher FRA than that of oxidized HDL_3_ at the 1:0.1, 1:0.5, and 1:1 molar ratio.

### 3.5. Protection of apoA-I and Enhancement of Antioxidant Activity by Co-Presence of CIGB-258

Figure 5A shows the distinct bands of apoA-I (28 kDa) and apoA-II (17 kDa) in HDL_2_ and HDL_3_ without the presence of any additional plasma proteins, such as albumin (67 kDa), suggesting the successful isolation of the purified form of HDL. As shown in Figure 5A, the co-addition of apoA-I:CIGB-258 (at molar ratio 1:1) into HDL resulted in more vigorous band intensity of apoA-I in both HDL_2_ and HDL_3_, which was 60% and 41%, respectively. In lane N, no lipid-free apoA-I was detected because the lipid-free apoA-I was not externally added. Interestingly, externally added lipid-free apoA-I (indicated by red arrowhead) was detected with the same band intensity under the apoA-I band in each HDL (indicated by black arrow). These results suggest that the elevations of the apoA-I band in both HDL_2_ and HDL_3_ were associated with a dose-dependent manner of CIGB-258 addition.

As shown in Figure 5B, the FRA of each HDL was enhanced by the addition of apoA-I:CIGB-258 with the dose-dependent manner of CIGB-258 up to 15~16% more increase at 1:1 molar ratio than that of an apoA-I alone (1:0) addition. Interestingly, an apoA-I alone addition did not enhance the FRA compared with HDL_2_ or HDL_3_ alone, suggesting that the presence of CIGB-258 substantially enhances the HDL-associated antioxidant ability via structural stabilization of apoA-I.

### 3.6. Enhanced Wound-Healing Activity by an Increase of CIGB-258

As shown in Figure 6A, the PBS alone group (photo a) showed the fastest wound healing during 48 h, while the CML+PBS group showed the slowest wound healing speed with the darkest methylene blue stained area. Interestingly, co-treatment of apoA-I and CIGB-258 improved the wound healing speed and morphology in a dose-dependent manner of CIGB-258, especially in a 1:1 molar ratio (photo f). As shown in Figure 6B, the PBS alone group and the CML+PBS group exhibited 67% and 14% wound healing, respectively, during 48 h-post treatment, suggesting the remarkable toxicity of CML to disturb the wound healing process. However, the co-addition of apoA-I:CIGB-258 ameliorated the CML-induced wound damage and improved the wound healing up to 35%, 37%, 53%, and 56% by 1:0, 1:0.1, 1:0.5, and 1:1 molar ratio, suggesting a dose-dependent manner of CIGB-258 to enhance the wound-healing activity.

### 3.7. Histological Analysis of the Wounded Area

The histology analysis of the wound site (skin plus muscle) was examined by the H&E staining, which revealed the formation of the epidermis (neo-optimization) across all the groups (Figure 7A). However, a fragmented and irregular epidermis formation was noticed in the only CML-treated wound, in contrast to an even and regular epidermis formation detected in the PBS control group. The CML-impaired neo-epithelization was substantially restored by the exposure of apoA-I:CIGB-258 in the concentration-dependent manner of CIGB-258. As depicted in Figure 7, a regular and even epidermis formation was observed in the 1:0.5 and 1:1 apoA-I:CIGB-258 treated group, which is somewhat similar to the effect observed in the PBS group, signifying the restoration of the wound site.

An enhanced granulation tissue formation (indicated by the red arrow) was observed in the PBS-treated group, which was significantly 2.6-fold (*p* < 0.001) higher than the thickness noticed in the treated group (Figure 7A,B). Likewise, a substantial curative effect of apoA-I:CIGB-258 at 1:0.1, 1:0.5, and 1:1 (molar ratio) was observed against CML-impaired granulation tissue formation. A significant 1.8-fold (*p* < 0.05), 2.0-fold (*p* < 0.01), and 2.4-fold (*p* < 0.001) higher granulation tissue formation was noticed in the apoA-I:CIGB258 at 1:0.1, 1:0.5, and 1:1 treated wound as compared to the only CML exposed wound signifying the potency of CIGB-258 towards CML impaired wound healing.

The DHE staining suggested a significant 4.5-fold (*p* < 0.001) higher ROS generation in the wounded site exposed to CML compared to the PBS control, indicating the higher oxidative stress environment in the CML-treated wound (Figure 7A,C). A 2.2-fold (*p* < 0.001), 4.0-fold (*p* < 0.001), and 4.5-fold (*p* < 0.001) lower DHE fluorescent intensity corresponding to the ROS production was observed in the wounded tissue treated with apoA-I:CIGB-258 at 1:0.1, 1:0.5 and 1:1, respectively, compared to the only CML treated wound highlighting the potential of CIGB-258 to inhibit CML induced oxidative stress and consequently wound healing.

Consistent with the DHE staining, the AO staining revealed a significantly lower apoptotic cell death in the wounded tissue treated with apoA-I:CIGB-258 than only the CML-treated wound (Figure 7A,C). A significant 2.3-fold (*p* < 0.01), 2.6-fold (*p* < 0.001), and 4.5-fold (*p* < 0.001) reduced AO fluorescent intensity corresponding to the apoptosis was noticed in apoA-I:CIGB-258 at 1:0.1, 1:0.5 and 1:1 treated wound, respectively, than the wound exposed to the CML only. Combined results explain the efficacy of CIGB-258 in inhibiting CML-posed ROS and apoptotic cell death, which leads to prompt wound recovery. 

### 3.8. Synergistic Anti-Inflammatory Activity of apoA-I and CIGB-258

As shown in Figure 8A, intraperitoneal injection of CML caused abrupt loss of swimming ability, all zebrafish lying down on the bottom of the tank via acute paralysis and convulsion in zebrafish, especially all zebrafish could not move at 30 min post-injection (photo b), while PBS-injected group showed normal and active swimming ability (photo a). The co-injection of apoA-I:CIGB-258 displayed the recovery of CML-impaired zebrafish swimming ability in a dose-dependent manner of CIGB-258 (photo c-f).

As shown in Figure 8B, the CML alone group showed 0% and 17% of swimming zebrafish at 30 min and 60 min post-injection, respectively, while the CML+apoA-I alone group showed that 10% and 33% of zebrafish recovered swimming ability at 30 min and 60 min post-injection. Interestingly, the co-addition of apoA-I:CIGB-258 induced remarkable recovery of swimming ability in a dose-dependent manner of CIGB-258. The proportion of swimming zebrafish at 30 min post-injecting apoA-I:CIGB-258 was 17% at a molar ratio of 1:0.1, 20% at a ratio of 1:0.5, and 33% at a ratio of 1:1. At 60 min post-injection, 40% (*p* < 0.01 vs. CML alone group), 47% (*p* < 0.001 vs. CML alone group), and 67% (*p* < 0.001 vs. CML alone group) of zebrafish restored their swimming ability in 1:0.1, 1:0.5, and 1:1 molar ratio of apoA-I:CIGB-258.

At 180 min-post injection, the CML+PBS group showed the lowest survivability, around 36%, while PBS alone group showed 100% survivability, indicating no damage occurred during intraperitoneal injection (Figure 8C). The co-injection of apoA-I:CIGB-258 elevated the survivability in a dose-dependent manner of CIGB-258. A 50%, 63% (*p* < 0.01), 67% (*p* < 0.001), and 83% (*p* < 0.001) survivability were observed in 1:0, 1:0.1, 1:0.5, and 1:1 molar ratio of the apoA-I:CIGB-258 treated zebrafish, respectively (Figure 8C).

### 3.9. Histological Examination of Hepatic Tissue

As shown in Figure 9A, hematoxylin & eosin (H&E) staining revealed that the CML+PBS group showed the largest stained area and strongest red intensity, suggesting that massive infiltration of neutrophils due to acute inflammation. While the PBS alone group showed the smallest H&E-stained area and the weakest red intensity, the co-injection of apoA-I:CIGB-258 induced a smaller stained area and less red intensity. Quantification of the H&E-stained area revealed that the PBS alone group showed the smallest area, around 2.9%, while the CML+PBS group showed the largest area, around 10.2% (Figure 9B). However, H&E-stained areas were reduced by the co-addition of apoA-I:CIGB-258 around 7.9%, 7.3% (*p* < 0.05), 6.6% (*p* < 0.05), and 4.1% (*p* < 0.01) at 1:0, 1:0.1, 1:0.5, and 1:1 molar ratio, respectively. As shown in Figure 9C, counting of infiltrated neutrophils revealed that the CML+PBS group showed the highest neutrophils, around 6.7-fold higher than the PBS alone group. Interestingly, the neutrophil numbers were reduced gradually by the co-addition of apoA-I:CIGB-258 in a dose-dependent manner of CIGB-258 as evidenced by 13%, 60% (*p* < 0.05), 71% (*p* < 0.05), and 88% (*p* < 0.01) reduced neutrophil counts, respectively, than that of CML+PBS group. These results suggest that the co-treatment of apoA-I and CIGB-258 ameliorated the acute inflammatory response induced by the exposure of CML in a dose-dependent manner of CIGB-258.

### 3.10. Production of Interleukin (IL)-6 in Liver

The hepatic IL-6 levels in response to CML and CIGB-258 treatment were determined by IHC staining. As depicted in Figure 10, a higher IL-6-stained area (40.4%) was observed in the CML-injected group that was significantly 9.0-fold (*p* < 0.001) higher than the IL-6-stained area quantified in the control (4.5%), suggesting the impact of CML on the IL-6 production. The CML-induced IL-6 production was effectively prevented by the treatment of apoA-I and apoA-I:CIGB-258. A significant 4.2-fold (*p* < 0.001), 5.2-fold (*p* < 0.001), and 10.6-fold (*p* < 0.001) reduced IL-6-stained area was quantified in the apoA-I mixed with CIGB-258 at the molar ratio of 1:0.1, 1:0.5, and 1:1 than the IL-6-stained area observed in the CML injected group, suggesting the effect of apoA-I and CIGB-258 to prevent CML induced IL-6 production.

### 3.11. Extent of Fatty Liver Change, Apoptosis, and ROS Production

As shown in Figure 11A, intraperitoneal injection of CML caused severe fatty liver change as indicated by the strongest red intensity in the oil red O staining. In contrast, the PBS alone group showed the weakest red intensity. As shown in Figure 11B, the CML+PBS group showed 2.5-fold higher fatty liver change than the PBS alone group, suggesting the fatty liver provocative role of CML. The co-treatment of apoA-I:CIGB-258 improved the fatty liver change evident by a 33% (*p* < 0.01), 43% (*p* < 0.001), 54% (*p* < 0.001), and 56% (*p* < 0.001) lower oil red O-stained area at 1:0, 1:0.1, 1:0.5, and 1:1 molar ratio, respectively, compared to the CML+PBS injected group.

AO and DHE staining revealed that the CML+PBS group exhibited the strongest green and red intensity (Figure 11A), suggesting severe apoptosis and ROS production, respectively. Quantification of fluorescence intensity (FI) revealed that the CML+PBS group showed 4.9-fold and 5.0-fold higher AO (green) and DHE (red) fluorescence, respectively, than those of the PBS alone group, indicating the impact of CML-to induce hepatic ROS generation and apoptosis (Figure 11C). The co-treatment of apoA-I:CIGB-258 reduced the AO and DHE fluorescence in a dose-dependent manner of CIGB-258, especially the 1:1 molar ratio injected group showed the lowest AO and DHE fluorescence, which was 77% (*p* < 0.001), and 70% (*p* < 0.001), less than those observed in CML+PBS injected group, respectively.

### 3.12. Histological Analysis of Kidney

The H&E staining, as depicted in Figure 12A, represents kidney morphology. Tightly packed distal tubules (DT) and proximal tubules (PT) were observed in the PBS control group, representing normal kidney morphology. In contrast, the kidney section from the CML-treated group showed disorganized, loosely arranged distal and proximal tubules with occasional cell debris in the tubular cast (indicated by the red arrow). Also, the prevalence of the basophilic cluster corresponds to the new nephron generation (indicated by green arrow), signifying the adverse effect of CML on the kidney. CML-induced kidney damage is substantially prevented by the treating apoA-I in combination with CIGB-258. To be precise, the apoA-I:CIGB-258 at the molar ratio 1:1 displayed a substantial recovery as evidenced by a compact arrangement of well-differentiated distal and proximal tubules with no visible sign of cell debris in the tubular cast.

The oil red O (ORO) staining displayed a significantly 2.1-fold (*p* < 0.001) higher accumulation of lipids in the kidney section of the CML-injected group as compared to the PBS-injected group, indicating the adverse impact of CML on the kidney (Figure 12A,B). Treatment of apoA-I:CIGB-258 at a molar ratio of 1:0.5 and 1:1 significantly prevented the CML-induced lipid accumulation as apparent by the significant 1.5-fold (*p* < 0.05) and 1.9-fold (*p* < 0.01) reduced ORO-stained area than the ORO stained quantified in the CML injected group. Contrary to this, no visible effect of apoA-I:CIGB-258 at the molar ratio 1:0.0 and 1:0.1 were observed on the CML-provoked lipid accumulation in the kidney.

The DHE staining revealed the visible effect of CML on ROS production in the kidney (Figure 12A, C). A significantly 1.9-fold (*p* < 0.05) higher DHE stained area corresponding to the ROS production was quantified in the CML-injected group compared to the PBS-injected control. Treatment of apoA-I with GIGB-258 at the molar ratio 1:0 and 1:0.1 displayed a non-significant (*p* > 0.05) effect against CML-induced ROS generation. However, the higher molar ratio of apoA-I: GIGB-258 (1:0.5 and 1:1) displayed a substantial impact to counter CML-induced ROS generation as manifested by a significant 1.5-fold (*p* < 0.05) and 3.5-fold (*p* < 0.01) reduced ROS production, respectively, as compared to the CML.

Consistent with the outcomes of DHE fluorescent staining, AO fluorescent staining revealed a higher prevalence of apoptosis in the CML injected group that was significantly 2.4-fold (*p* < 0.05) and 5.2-fold (*p* < 0.01) reduced following the treatment of apoA-I:GIGB-258 at the molar ration of 1:0.5 and 1:1; testifying the anti-apoptotic role of apoA-I in combination with GIGB-258. Combined results of kidney histology establish the influential role of apoA-I and GIGB-258 in preventing kidney morphology impaired by the CML-provoked lipid accumulation, oxidative stress, and apoptosis.

### 3.13. Change of Plasma Lipid Profile

As shown in Figure 13A,B, the IP injection of CML caused an acute increase of plasma total cholesterol (TC) and triglyceride (TG), which was 1.6- and 1.7-fold higher than that quantified in the PBS alone group. The CML elevated plasma TC level was significantly decreased by the co-treatment of apoA-I:CIGB-258 in a dose-dependent manner. At the apoA-I:CIGB-258 molar ratio 1:0, 1:0.1, 1:0.5, and 1:1, a 23% (*p* < 0.01), 32% (*p* < 0.01), 30%(*p* < 0.01), and 36% (*p* < 0.001) lower TC was quantified as compared to the CML+PBS group, respectively. Similarly, the apoA-I:CIGB-258 dose-dependent alleviated the CML-induced plasma TG level. At the molar ratio of 1:0, 1:0.1, 1:0.5, and 1:1 of apoA-I:CIGB-258 a 23% (*p* < 0.05), 26% (*p* < 0.05), 40% (*p* < 0.001), and 47% (*p* < 0.001) lower TG was detected compared to the CML+PBS group. Unlike the TC, the CML-induced TG level decreased linearly with the increasing proportion of CIGB-258.

Consistent with the findings of TC and TG, the combination of apoA-I and CIGB-258 displayed a substantial effect on elevating the HDL-C level impaired by CML (Figure 13C). At a 1:1 molar concentration of apoA-I and CIGB-258, the HDL-C level was detected to be substantially 2-fold and 3-fold higher compared to the HDL-C levels observed in apoA-I alone (1:0 molar ratio) and CML-injected groups, respectively, testifying to the efficacy of CIGB-258 in elevating plasma HDL-C level.

### 3.14. Change of Plasma Hepatic Enzyme Levels

As shown in Figure 14, CML treatment caused acute elevation of hepatic enzyme levels in plasma. CML+PBS group showed 1.7- (*p* < 0.001) and 7.8-fold (*p* < 0.001) higher plasma AST and ALT, respectively, than those of PBS alone group. The CML elevated plasma AST level was significantly decreased by the co-treatment of apoA-I:CIGB-258 up to 23% (*p* < 0.001), 25% (*p* < 0.001), 32% (*p* < 0.001), and 40% (*p* < 0.001) at 1:0, 1:0.1, 1:0.5, and 1:1 molar ratio, respectively. Notably, the co-treatment with apoA-I and CIGB-258 resulted in a more gradual and significant reduction in plasma ALT levels. A 42% (*p* < 0.01), 71% (*p* < 0.001), 73% (*p* < 0.001), and 87% (*p* < 0.001) lower ALT level were quantified in the groups treated with apoA-I:CIGB-258 administered at 1:0, 1:0.1, 1:0.5, and 1:1 molar ratio, respectively, compared to the CML+PBS injected group. These findings indicate that plasma AST and ALT levels, which were elevated due to CML-induced inflammation, were substantially reduced with the treatment of apoA-I and CIGB-258.

## 4. Discussion

In a lipid-free state, the apoA-I:CIGB-258 mixture showed synergistic antioxidant activity with increased CIGB-258 content to protect HDL from cupric ion-mediated oxidation. The oxidized HDL showed degradation of apoA-I and aggregated HDL morphology with smaller particle size (photo b, Figure 2 and Figure 4). However, the co-treatment of apoA-I:CIGB-258 improved the HDL particle morphology with a more distinct particle shape and bigger size, indicating that the CIGB-258 could improve HDL particle quality. As well as lowering oxidized species in HDL (Figure 1B and Figure 3B), the apoA-I band increased, and the ferric ion reduction ability was also elevated by the increase of CIGB-258 content (Figure 5). The increase in CIGB-258 content accelerated the wound-healing activity of the apoA-I:CIGB-258 mixture with the least production of ROS and apoptosis under the presence of CML (Figure 6 and Figure 7). The superior wound healing observed with increasing amounts of CIGB-258 in apoA-I:CIGB-258 formulations may be due to CIGB-258, a peptide originating from HSP60. This finding aligns with an earlier report highlighting the effectiveness of extracellular HSP60 in prompting wound healing [46].

Under the acute pro-inflammatory phase, an increase in the CIGB-258 content in the apoA-I:CIGB-258 mixture contributes more to protecting zebrafish from acute paralysis and death (Figure 8), lowering inflammation in the liver (Figure 9 and Figure 10) and kidney (Figure 12) and improving the blood-lipid profile (Figure 13) with lower AST/ALT (Figure 14).

Although many pharmaceutical approaches have focused on raising HDL-C, such as fibrates, niacin, and CETP inhibitors (i.e., Torcetrapib, Evacetrapib, Anacetrapib), they were not successful in marketing due to adverse effects and off-target interactions [47,48,49]. Moreover, intravenous infusions of HDL and apoA-I (MDCO-216, CER-001, CSL111, and CSL112) were also disappointingly associated with HDL-C raising therapies despite their relatively higher dosage of around 3–80 mg/kg [50,51,52,53]. However, no successful pharmaceutical agent has been able to exert an anti-inflammatory effect to reduce CRP, IL-6, and TNF-α via the stabilization of apoA-I and HDL. As far as the authors’ knowledge, there has been no drug reported to stabilize HDL and apoA-I via protection of protein structure; CIGB-258 is the first peptide drug candidate to stabilize apoA-I structure to suppress the cytokine cascade, especially the production of IL-6.

More interestingly, the oxidatively modified apoA-I can be attached in the electronegative LDL with amino acid modification, suggesting that oxidation of apoA-I severely occurred with proteolytic degradation simultaneously with oxidation of LDL [54]. SARS-CoV-2 infection triggers pro-atherogenic inflammatory responses to upregulate the CD36 receptor gene for oxLDL and downregulate the cholesterol efflux gene for ABCA1 [55]. Consequently, several pro-inflammatory pro-atherogenic cytokines, IL-6, IL-1β, and interferon-α, were released by both infected macrophages and foam cells to induce cytokine storm. HDL inhibits LDL oxidation through antioxidants apoA-I by suppressing adhesion molecules and monocyte chemoattractant protein (MCP)-1. HDL also attenuates the stimulatory effects of oxLDL on macrophages to release inflammatory cytokines [56,57]. However, it has been well known that SARS-CoV-2 infection is directly associated with the generation of dysfunctional HDL, which could exacerbate pro-inflammatory signaling and immune disorders. The induction of dysfunctional HDL was initiated by modification of apoA-I, such as oxidation, myeloperoxidase, and glycation, to impair cholesterol efflux activity via fragmentation and degradation of apoA-I [58,59]. Therefore, prevention of oxidation and glycation of apoA-I by co-presence of CIGB-258 might contribute to preserving the apoA-I structure and HDL functionality to yield the anti-infectious and anti-inflammatory activities similar to IL-6 inhibitor, tocilizumab [17,21,22,25].

Similarly to glycation stress, CML also negatively impacts the structure and function of apoA-I/HDL. Intraperitoneal administration of CML in zebrafish resulted in acute paralysis, impaired swimming ability, and a sharp increase in neutrophil infiltration, IL-6, and TNF-α levels [25], resembling the cytokine storm observed in COVID-19 patients. However, co-treatment of CIGB-258 recovered swimming ability and suppressed IL-6 and TNF-α in the zebrafish via protection of apoA-I and HDL in the dose-dependent manner of CIGB-258. These findings make a strong agreement with the previous clinical findings that intravenous injection of CIGB-258 (1–2 mg) suppressed cytokine cascade in hyperinflammatory COVID-19 patients via controlling neutrophil activation with lowering IL-6, IL-10, TNF-α, and CRP, and raising regulatory T cells (T_reg_) [16]. As well as in rheumatoid arthritis (RA) animal models and patients, CIGB-258 increased the frequency of T_reg_ with a reduction of proinflammatory cytokines [60,61]. These HDL/apoA-I protection effects of CIGB-258 were linked with: (1) a decrease of neutrophil activation, (2) a decrease of proinflammatory cytokine levels, (3) a decrease of granzyme B and perforin, (4) an increase of regulatory T cells, and (5) without causing immunosuppression. These mechanistic insights of CIGB-258 may apply to the treatment of patients with sepsis because low HDL-C levels in serum and displacing apoA-I from HDL by serum amyloid A (SAA) are major risk factors for sepsis. HDL is well known for the neutralizing effect of lipopolysaccharide (LPS), which can bind to a toll-like receptor (TLR)-4 to initiate a signaling cascade [62]. ApoA-I protects from sepsis by binding to lipopolysaccharide (LPS) [63,64]. The protective effect of CIGB-258 for apoA-I could prevent the production of proinflammatory cytokines in neutrophils, vascular endothelial dysfunction, and organ injury in sepsis.

## 5. Conclusions

In vitro and in vivo evaluation revealed that apoA-I and CIGB-258, in their lipid-free form, display synergistic antioxidant and anti-inflammatory effects against CML-induced acute inflammation. With increasing CIGB-258 content in a mixture of apoA-I:CIGB-258, from 1:0, 1:0.1, 1:0.5, and 1:1, both HDL_2_ and HDL_3_ were protected from oxidation to maintain their structural stability and antioxidant abilities. The dermal application of apoA-I:CIGB-258 mixture showed synergistic wound healing function with the least ROS production and apoptosis caused by CML compared to the highest content of CIGB-258. The intraperitoneal injection of apoA-I:CIGB-258 mixture protected CML-induced acute death of zebrafish with the least neutrophil infiltration and IL-6 production in the liver links to the highest content of CIGB-258. In summary, CIGB-258’s ability to protect apoA-I/HDL led to superior wound healing, increased zebrafish survival rates, and improved antioxidant capabilities.

## Figures and Tables

**Figure 1 antioxidants-13-01049-f001:**
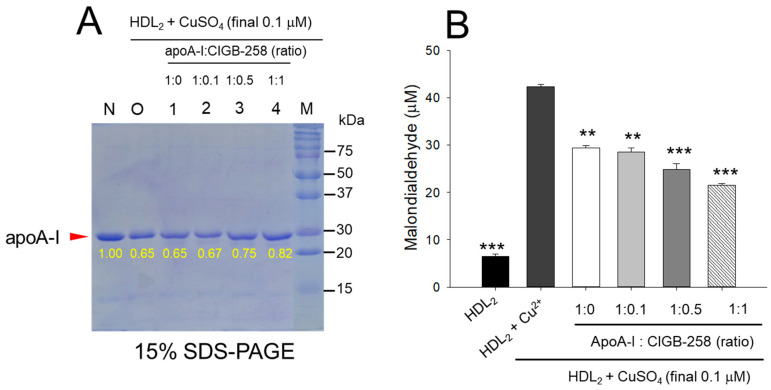
Protection of HDL_2_ from cupric ion mediated oxidation by apoA-I:CIGB-258 content. (**A**) Electrophoretic profile (in 15% SDS-PAGE) of HDL_2_ under the presence of cupric ion and mixture of apoA-I:CIGB-258 at different molar ratios. (**B**) Quantification of oxidized species examined by TBARS assay in the HDL_2_ treated with cupric ions in the presence and absence of apoA-I:CIGB-258 at different molar ratios. The ** (*p* < 0.01), and *** (*p* < 0.001) denote significant distinctions between the groups compared to the HDL_2_+cupric ion (Cu^2+^) group.

**Figure 2 antioxidants-13-01049-f002:**
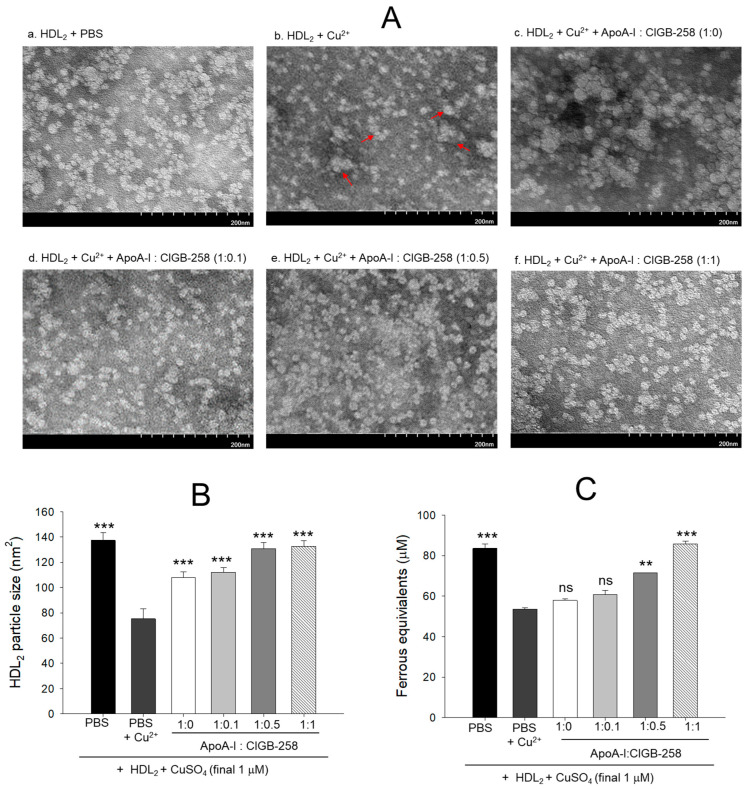
Protection of HDL_2_ from cupric ion mediated oxidation by co-presence of apoA-I and CIGB-258. (**A**) Transmitted electron microscopy (TEM) image analysis of HDL_2_ with various ratios of apoA-I and CIGB-258 under the presence of cupric ion. The red arrow indicates aggregated HDL_2_ by oxidation. (**B**) Measurement of the HDL_2_ particle size using Hitachi EMIP-EX software (version 07.13) for TEM images obtained from the different groups. (**C**) Ferric ion reduction ability of HDL_2_ under the presence of cupric ion and mixture of apoA-I:CIGB-258. The ** (*p* < 0.01), and *** (*p* < 0.001) denote significant distinctions between the groups compared to the HDL_2_ + cupric ion (Cu^2+^) group. The ns represent the non-significant difference between the groups.

**Figure 3 antioxidants-13-01049-f003:**
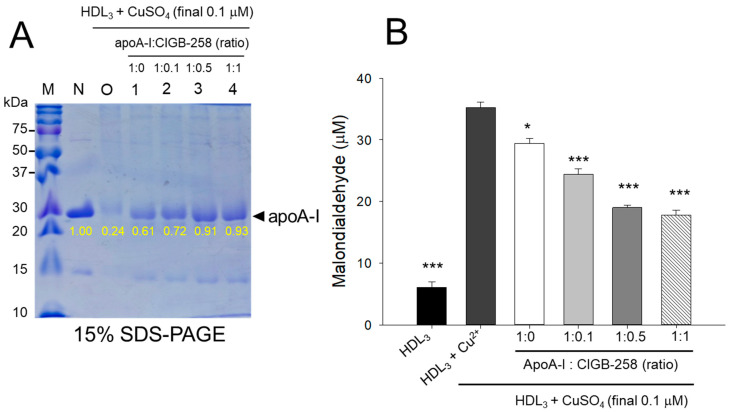
Protection of HDL_3_ from cupric ion mediated oxidation by apoA-I:CIGB-258 content. (**A**) Electrophoretic profile (in 15% SDS-PAGE) of HDL_3_ under the presence of cupric ion and mixture of apoA-I:CIGB-258 at different molar ratios. (**B**) Quantification of oxidized species examined by TBARS assay in the HDL_3_ treated with cupric ions in the presence and absence of apoA-I:CIGB-258 at different molar ratios. The * (*p* < 0.05), and *** (*p* < 0.001) denote significant distinctions between the groups compared to the HDL_3_+cupric ion (Cu^2+^) group.

**Figure 4 antioxidants-13-01049-f004:**
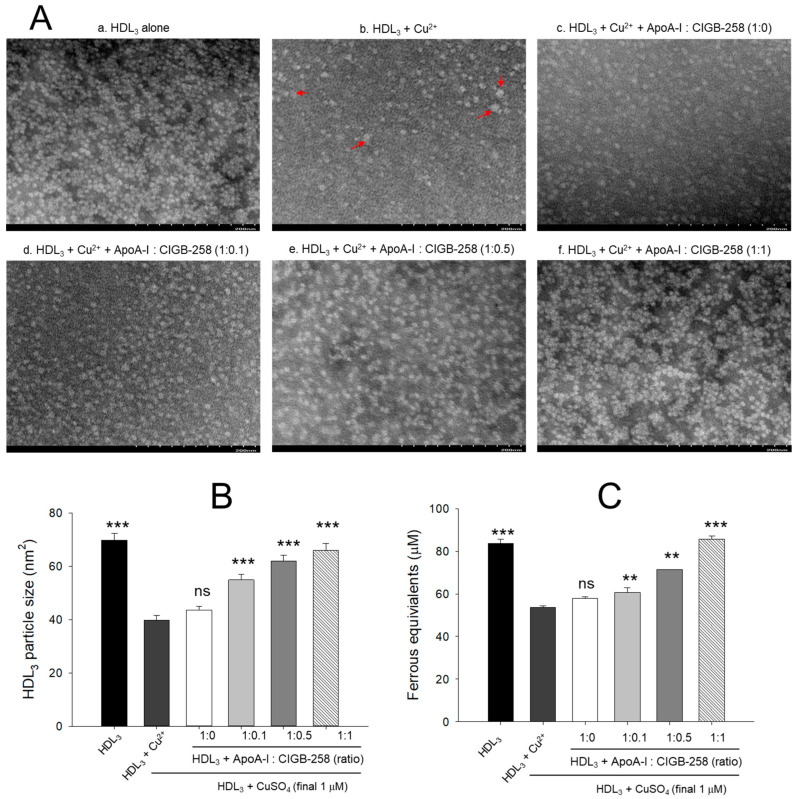
Protection of HDL_3_ from cupric ion mediated oxidation by co-presence of apoA-I and CIGB-258. (**A**) Transmitted electron microscopy (TEM) image analysis of HDL_3_ with various ratios of apoA-I and CIGB-258 under the presence of cupric ion. The red arrow indicates aggregated HDL_3_ by oxidation. (**B**) Measurement of the HDL_3_ particle size using Hitachi EMIP-EX software (version 07.13) for TEM images obtained from the different groups. (**C**) Ferric ion reduction ability of HDL_3_ under the presence of cupric ion and mixture of apoA-I:CIGB-258. The ** (*p* < 0.01), and *** (*p* < 0.001) denote significant distinctions between the groups compared to the HDL_3_+cupric ion (Cu^2+^) group. The ns represent the non-significant difference between the groups.

**Figure 5 antioxidants-13-01049-f005:**
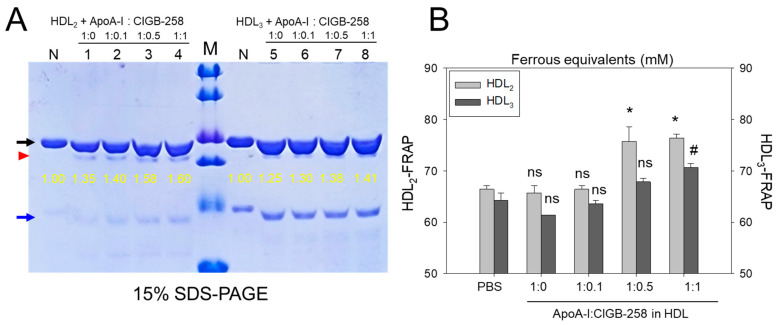
Effect of apoA-I in the presence of CIGB-258 on the stabilization of HDL-associated apoA-I and the antioxidant activity. (**A**) Electrophoretic profiles of HDL_2_ (1 mg/mL, 50 μg) and HDL_3_ (1 mg/mL, 50 μg) under co-presence of apoA-I:CIGB-258 at molar ratio of 1:0, 1:0.1, 1:0.5 and 1:1. The black and blue arrow indicates apoA-I (28 kDa) and apoA-II (17 kDa) in each HDL. The yellow numbers indicate the band intensity of apoA-I in each HDL. The red arrowhead indicates externally added lipid-free apoA-I. Lane M, Precision Plus Protein^TM^ Standards (BioRad, Hercules, CA, USA #161-0374) containing 75, 50, 37, 25, 20, and 15 kDa from the top position. (**B**) Measurement of paraoxonase activity with HDL_2_ and HDL_3_ under co-presence of apoA-I and CIGB-258 at 1:0.1, 1:0.5, and 1:1 molar ratio. The *, and # denotes significant distinctions (*p* < 0.05) for FRA in HDL_2_ and HDL_3_ groups, respectively, compared to the PBS+HDL_2_ or HDL_3_ group. The ns represent the non-significant difference between the groups.

**Figure 6 antioxidants-13-01049-f006:**
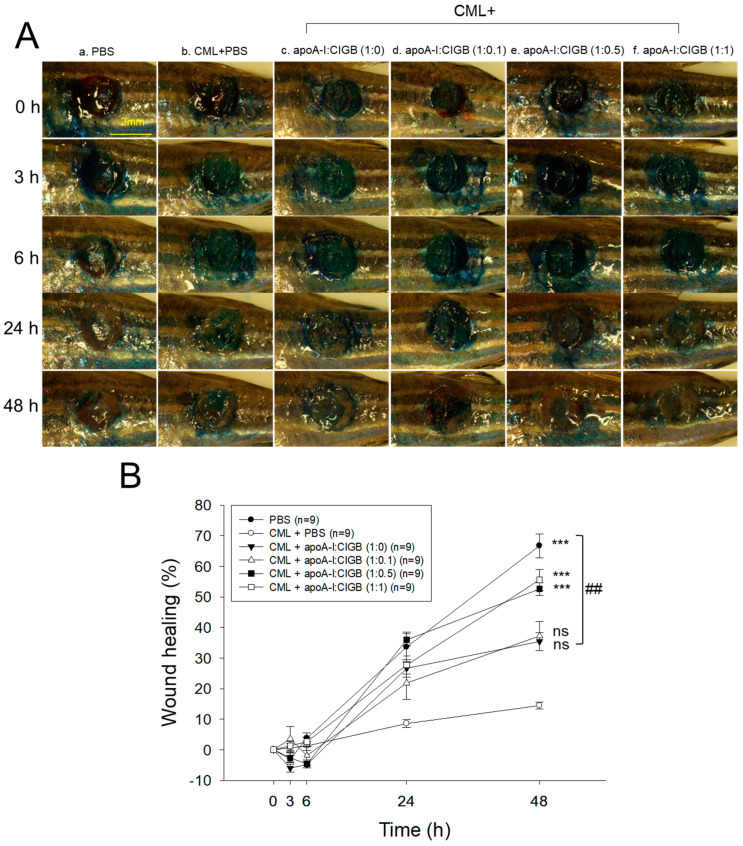
Wound healing activity of apoA-I combined with CIGB-258 in the presence of carboxymethyllysine (CML) in zebrafish. (**A**) Representative image of wounded site stained with methylene blue (final 0.1%) during 48 h post-treatment. (**B**) Time-dependent wound healing progression (percentage) during 48 h post-treatment; the wound healing percentage was calculated by comparing the stained area at a distinct time point with the initial wound area at 0 h. The *** (*p* < 0.001) denotes significant distinctions between the groups compared to the CML-treated group while ## depicts significant difference compared to the CML+apoA-I:CIGB-258 (1:0)-treated group. The ns represent the non-significant difference between the groups.

**Figure 7 antioxidants-13-01049-f007:**
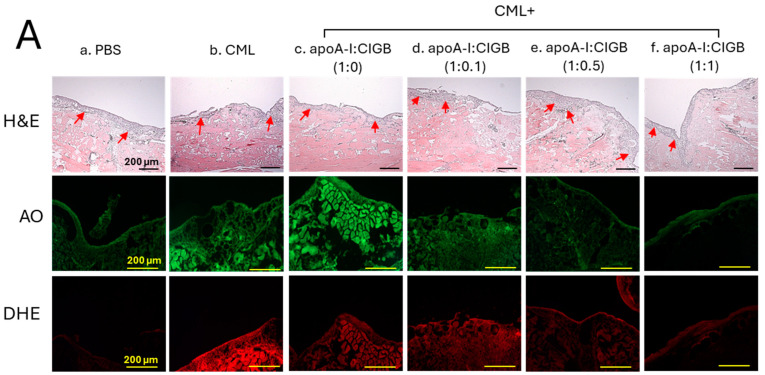
The histology of carboxymethyllysine (CML) impaired wound treated with apoA-I and CIGB-258. Treatment with apoA-I and CIGB-258 inhibits the effect of CML on wound healing. (**A**) Hematoxylin and eosin (H&E) staining, acridine orange (AO), and dihydroethidium (DHE) fluorescent staining representing morphology, reactive oxygen species (ROS), and extent of apoptosis, respectively, at 48 h post-treatment; images are 400× magnified, scale bar 100 μm. The red arrow in the H&E-stained images indicated granulation. (**B**) Average thickness of granulation. (**C**) The AO and DHE, fluorescent intensity quantification, was determined by Image J software (version 1.53r, http://rsb.info.nih.gov/ij, accessed on 16 January 2023). The *, **, and *** represent the statistical difference at *p* < 0.05, *p* < 0.01, and *p* < 0.001, compared to the CML treated group (for granulation thickness and AO fluorescent intensity), while ### represents the statistical difference at *p* < 0.001 compared to the CML treated group for DHE stained area; ns represent the non-significant difference between the groups.

**Figure 8 antioxidants-13-01049-f008:**
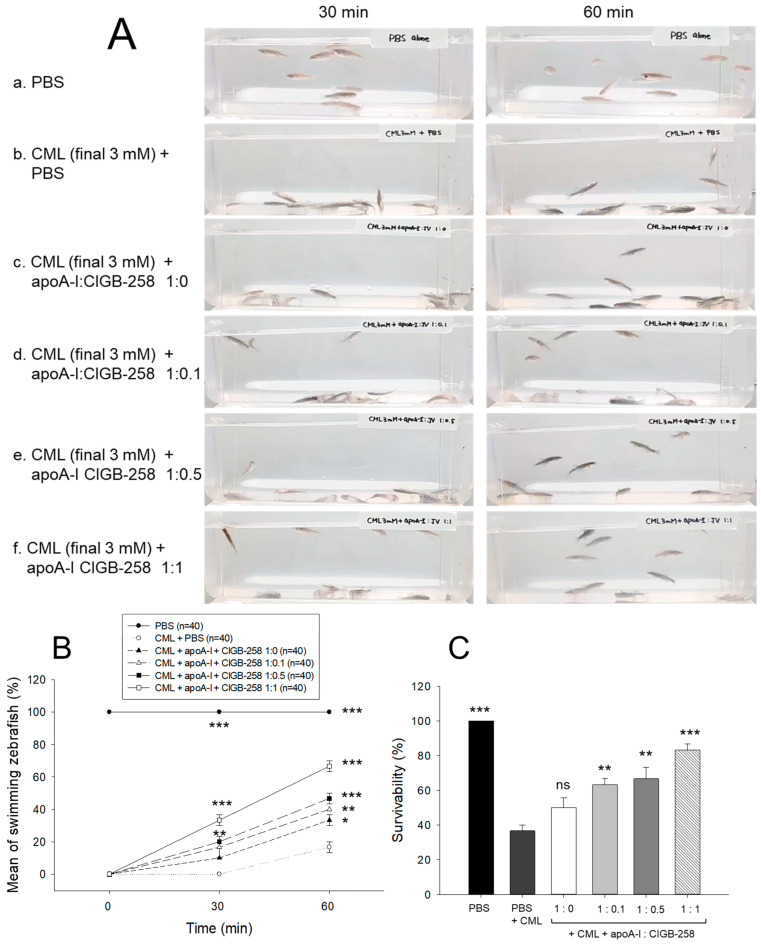
Comparison of swimming ability and survivability of zebrafish in the presence of carboxymethyllysine (CML, final 250 μg) and apoA-I and CIGB-258 with different molar ratio, 1:0, 1:0.1, 1:0.5, and 1:1. (**A**) Still image of swimming pattern at 30 min and 60 min post-treatment of CML and apoA-I:CIGB-258. (**B**) Mean of swimming activity at 30 min and 60 min post-treatment of CML and apoA-I:CIGB-258. (**C**) Survivability of zebrafish at 180 min post-treatment of CML and apoA-I:CIGB-258. The *, **, and *** represent the statistical difference at *p* < 0.05, *p* < 0.01, and *p* < 0.001, compared to the CML-treated group. The ns represent the non-significant difference between the groups.

**Figure 9 antioxidants-13-01049-f009:**
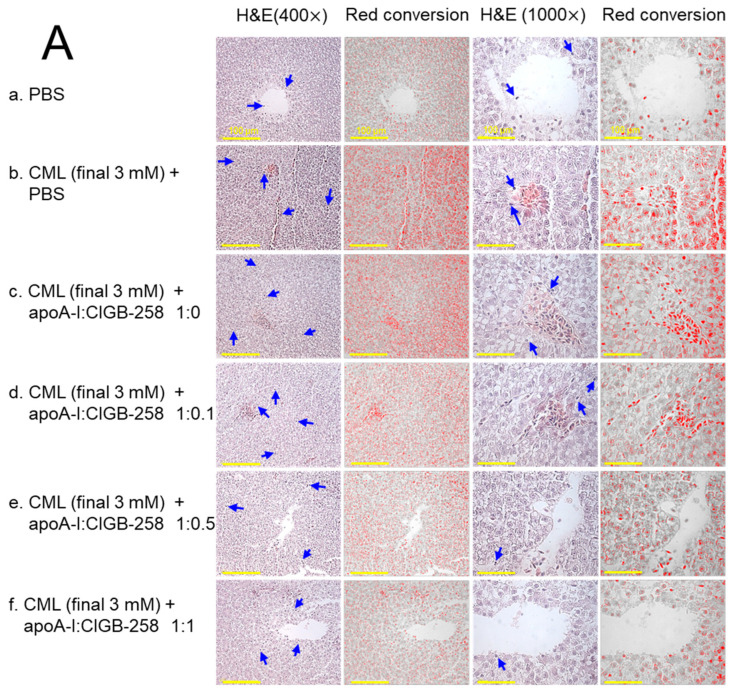
The effect of apoA-I with CIGB-258 on the carboxymethyllysine (CML) induced hepatic damage of adult zebrafish. (**A**) Hematoxylin and eosin (H&E) staining; images are presented as 400× and 1000× magnification. The blue arrow represents neutrophil infiltration. To enhance the visualization, the blue color of the H&E-stained area was interchanged with red color (red conversion) employing Image J software (version 1.53r, assessed on 16 January 2023) at a threshold value of 0–120 [scale bar, 100 μm]. (**B**) Quantification of the H&E-stained area across the different groups. (**C**) Neutrophil counts were assessed through microscopic investigation of a designated 1.23 mm^2^ area across distinct sections (n = 6), with five different regions analyzed within each section. The *, **, and *** represent the statistical difference at *p* < 0.05, *p* < 0.01, and *p* < 0.001, compared to the CML-treated group; ns represent non-significant differences between the groups.

**Figure 10 antioxidants-13-01049-f010:**
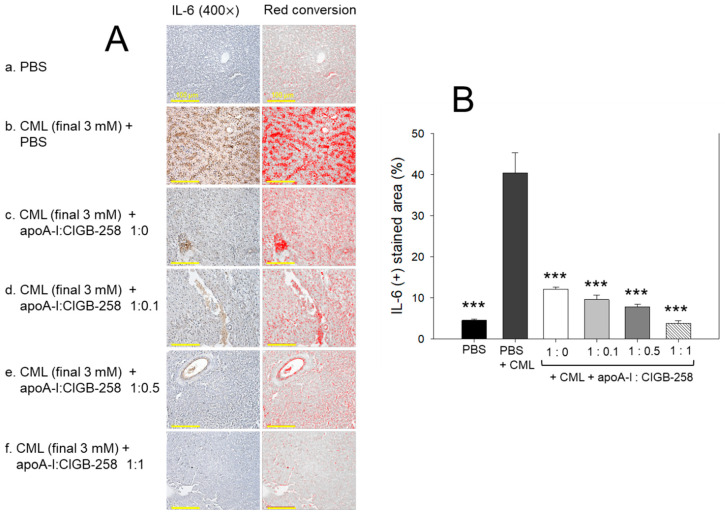
Immunohistochemical (IHC) analysis for the detection of interleukin (IL)-6 level in the hepatic tissue of carboxymethyllysine (CML) and apoA-I with CIGB injected adult zebrafish. (**A**) IHC images captured at 400× magnification [scale bar, 100 μm]. A red conversion of the IHC stained area (brown color) was performed utilizing Image J software (version 1.53r, http://rsb.info.nih.gov/ij accessed on 16 January 2023) at the brow color threshold value 20–100 to enhance the visibility of the immuno-stained area. (**B**) Image J software-based (version 1.53r, http://rsb.info.nih.gov/ij accessed on 16 January 2023) quantification of the IL-6-stained area. The *** represents the statistical difference at *p* < 0.001 compared to the CML-treated group.

**Figure 11 antioxidants-13-01049-f011:**
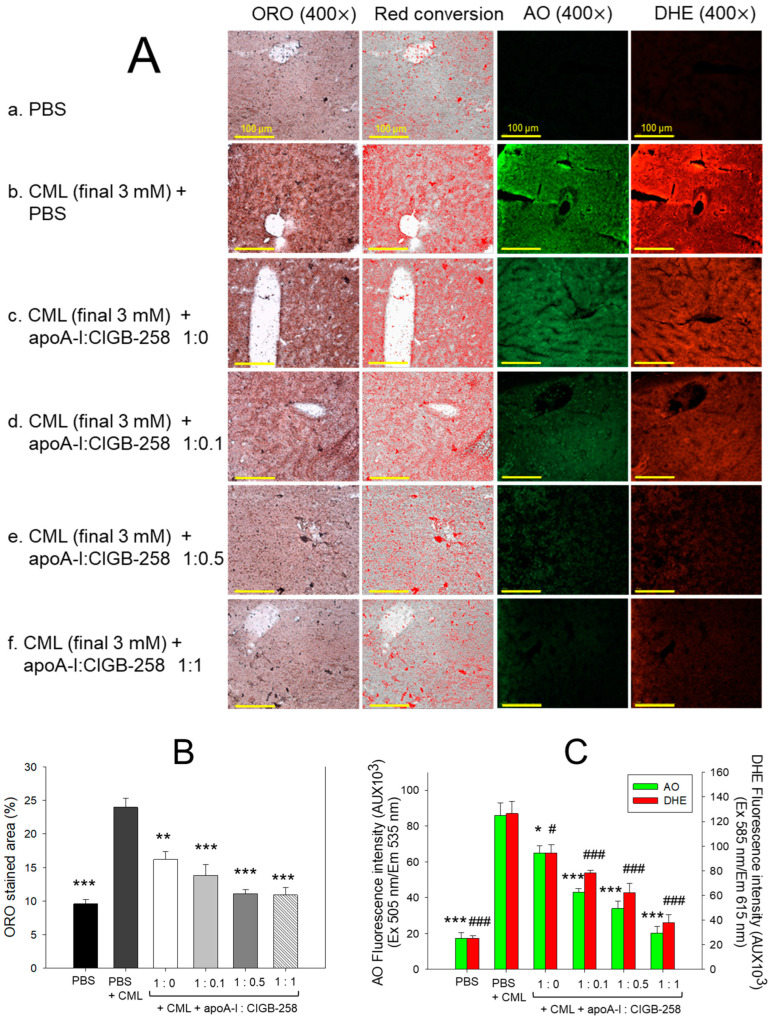
The effect of apoA-I and CIGB-258 treatment at different molar ratios on the carboxymethyllysine (CML) induced fatty liver change, apoptosis, and production of reactive oxygen species in hepatic tissue. (**A**) Hepatic images of oil red O stained area; image J software (version 1.53r, http://rsb.info.nih.gov/ij, accessed on 16 January 2023) based red conversion of ORO stained area to enhance the visualization; acridine orange (AO), and dihydroethidium (DHE) stained area. (**B**) Quantification of oil red O-stained area. (**C**) Quantification of AO and DHE fluorescent intensity corresponds to apoptosis and ROS production. The *, **, and *** represent the statistical difference at *p* < 0.05, *p* < 0.01, and *p* < 0.001, compared to the CML-treated group (for ORO-stained area and AO fluorescent intensity), while # and ### represent the statistical difference at *p* < 0.05 and *p* < 0.001 compared to the CML-treated group for DHE fluorescent intensity.

**Figure 12 antioxidants-13-01049-f012:**
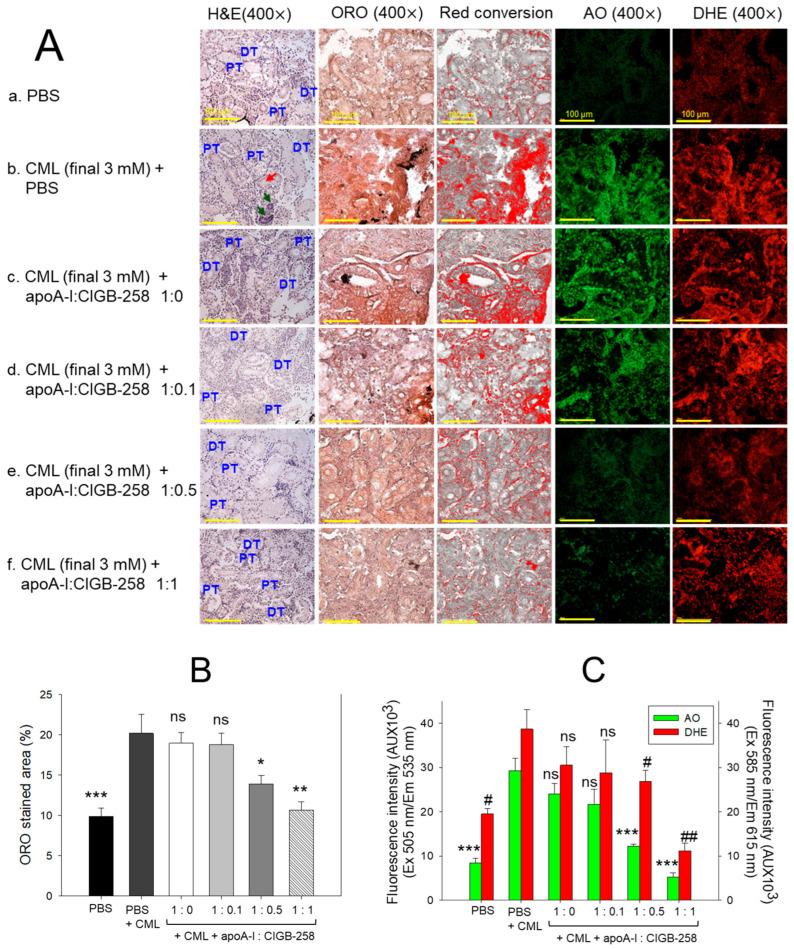
Effect of apoA-I with CIGB-258 on the carboxymethyllysine (CML) induced kidney damage of adult zebrafish. (**A**) Hematoxylin and eosin (H&E) staining; oil red O (ORO) staining. The ORO-stained area (brown color) was interchanged with red color using Image J software (version 1.53r, http://rsb.info.nih.gov/ij, accessed on 16 January 2023), acridine (AO), and dihydroethidium (DHE) fluorescent staining. The DT and PT are abbreviations for the distal tubules and proximal tubules; the red arrow indicates the luminal debris in the tubular cast, while the green arrow represents the basophilic cluster corresponding to new nephrons. All the images are captured at 400× magnification [scale bar, 100 μm]. (**B**) Quantification of the ORO-stained area. (**C**) Image J-based (version 1.53r, http://rsb.info.nih.gov/ij, accessed on 16 January 2023) quantification of AO and DHE fluorescent intensity representing the reactive oxygen species (ROS) and extent of apoptosis. The *, **, and *** represent the statistical difference at *p* < 0.05, *p* < 0.01, and *p* < 0.001, compared to the CML treated group (for ORO-stained area and AO fluorescent intensity), while # and ## represent the statistical difference at *p* < 0.05 and *p* < 0.001 compared to the CML treated group for DHE stained area; ns represent the non-significant difference between the groups.

**Figure 13 antioxidants-13-01049-f013:**
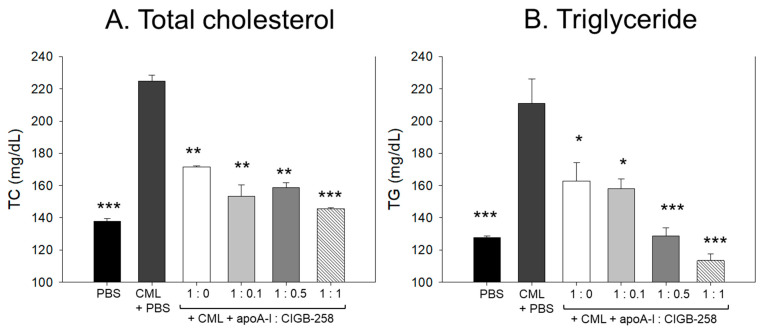
Effect of varied proportion of apoA-I and CIGB-258 on the plasma lipid profile of CML impaired zebrafish. (**A**,**B**) Total plasma cholesterol and triglycerides, respectively. (**C**,**D**) Plasma HDL-C and HDL-C/TC levels, respectively. The *, **, *** represents the statistical difference at *p* < 0.05, *p* < 0.01 and *p* < 0.001 compared to the CML-treated group.

**Figure 14 antioxidants-13-01049-f014:**
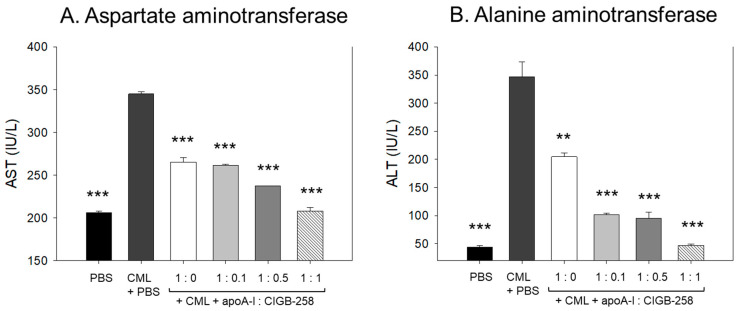
Quantification of hepatic enzymes in zebrafish plasma to compare the extent of hepatic damage, which was induced by intraperitoneal injection of carboxymethyllysine. (**A**) Comparison of plasma aspartate aminotransferase (AST) in zebrafish treated CML along with apoA-I: CIGB-258. (**B**) Comparison of alanine aminotransferase (ALT) in the zebrafish plasma. The **, *** represents the statistical difference at *p* < 0.01 and *p* < 0.001 compared to the CML-treated group.

## Data Availability

The data used to support the findings of this study are available from the corresponding author upon reasonable request.

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
