# Peer review of "Enhancing Wound Healing and Anti-Inflammatory Effects by Combination of CIGB-258 and Apolipoprotein A-I against Carboxymethyllysine Toxicity in Zebrafish: Insights into Structural Stabilization and Antioxidant Properties"

_antioxidants, 2024, doi:10.3390/antiox13091049_

Round 1

Reviewer 1 Report

Expanding on the insights of the earlier research, the current study investigated the synergistic interaction of apoA-I:CIGB-258 in safeguarding human HDL2 and HDL3 against oxidative stress, focusing on improvement in particle morphology, shape, size, and ferric ion reduction capacity. Additionally, physiological assessments were conducted to evaluate the wound healing and anti-inflammatory activities of the apoA- I:CIGB-258 mixture in the presence of CML in adult zebrafish.

The manuscript is well written, well organized and easy to read. The figures are well done and the authors have taken good care of the presentation. As a result, the possibility of the manuscript being published in this journal is strengthened. The only thing that can be improved is the part dedicated to wound healing that is not described by the authors and should be added in the introduction. Furthermore, since the Zebrafish model is introduced, it would be useful to talk about it a little and describe (if there are) examples of the reaction of this fish to an insult caused by a wound.

Author Response

Thank you for your insightful comments. Following the reviewer’s suggestion, we made point-to-point response and reflected on revision.

Please find attached doc as our response

Reviewer 2 Report

This study revealed the synergistic antioxidant and anti-inflammatory effects of apoA-I and CIGB-258 in the absence of phospholipids. However, there are a few minor issues that need to be corrected before the paper can be published.

1. Abstract. Results showed significant differences should add P< 0.05.

2. Introduction. Is there a specific link between high-density lipoproteins (HDL) and apolipoprotein A-I (apoA-I) and do they interact? What about LDL? Whether only one apolipoprotein binds specifically to HDL.

3. Line 58. Please add whether this small peptide has been studied in other experimental animals. e.g. model organisms.

4. Line 67. What are the consequences of intrinsic tryptophan (Trp) exposure? Do glycogenic amino acids promote gluconeogenesis?

5. Line 83. Is there a difference in function between the two subtypes of HDL?

6. Line 140. Please add what strain of zebrafish.

7. Results. Significant results in the data are identified with a p-value at the end of the sentence. Like p<0.01.

8. Fig.1A. The stripes are fuzzy. Isn't there a more optimised image?

Fig.1B. apoA-I:CIGB-258 mixture equal 1:0 Why are there no error bars in the chart notes? Please check others Fig.

9. Fig.7. Please standardise the identification of the prominence of the figure notes and adjust the font size

10.Line 665. How to exclude impaired motility or even indirect fish mortality due to handling of the injections

11. There are some grammar mistakes in the MS. Please check the English writing carefully.

Author Response

(The authors gave the same response as above.)
